# Antiviral Drugs in Influenza

**DOI:** 10.3390/ijerph19053018

**Published:** 2022-03-04

**Authors:** Magdalena Świerczyńska, Dagmara M. Mirowska-Guzel, Edyta Pindelska

**Affiliations:** 1Centre for Preclinical Research and Technology CePT, Department of Experimental and Clinical Pharmacology, Medical University of Warsaw, Banacha 1B, 02-097 Warsaw, Poland; lek.wet.magdaswierczynska@gmail.com; 2Department of Analytical Chemistry and Biomaterials, Faculty of Pharmacy, Medical University of Warsaw, Banacha 1B, 02-093 Warsaw, Poland; edyta.pindelska@wum.edu.pl

**Keywords:** influenza treatment, amantadine, neuraminidase inhibitors, zanamivir, oseltamivir, peramivir, laninamivir, baloxavir marboxil

## Abstract

Flu is a serious health, medical, and economic problem, but no therapy is yet available that has satisfactory results and reduces the occurrence of these problems. Nearly 20 years after the registration of the previous therapy, baloxavir marboxil, a drug with a new mechanism of action, recently appeared on the market. This is a promising step in the fight against the influenza virus. This article presents the possibilities of using all available antiviral drugs specific for influenza A and B. We compare all currently recommended anti-influenza medications, considering their mechanisms of action, administration, indications, target groups, effectiveness, and safety profiles. We demonstrate that baloxavir marboxil presents a similar safety and efficacy profile to those of drugs already used in the treatment of influenza. Further research on combination therapy is highly recommended and may have promising results.

## 1. Introduction

Influenza is a contagious disease that affects populations on a global scale. In humans, the etiological factor causing infections is influenza A or B virus and, to a small extent, influenza C virus. However, the greatest threat to humans is the type A virus due to its strong tendency toward antigenic variation and its pandemic potential. The disease can run its course in various ways, from asymptomatic infections or a mild infection of the upper respiratory tract to severe disease with high fever, chills, muscle pain, pneumonia, and even death [1]. According to the World Health Organization (WHO), the annual global influenza attack rate ranges from 20% to 30% of the child population and up to 10% of the adult population [2]. The number of deaths due to influenza is approximately 290,000–650,000 annually [3]. These factors constitute a global health, medical, and economic burden [4,5,6].

To reduce the influenza problem, research is constantly being carried out to monitor the directions of antigenic changes in the influenza virus. There is also ongoing work on new antiviral drugs and vaccines, the compositions of which are reconfigured every year [1]. Nevertheless, treating influenza remains a challenge, and the selection of appropriate drugs and the potential of combination therapy require a thorough knowledge of medicines available on the market and consideration of associated factors, such as the patient’s age, general health, and increased risk of possible complications. There are currently three registered drug types that specifically target the influenza virus: M2 proton channel antagonists (amantadine), neuraminidase inhibitors (NAIs; zanamivir, oseltamivir, peramivir, and laninamivir), and polymerase acidic endonuclease inhibitor (baloxavir marboxil), which is new in terms of its mechanism of action [7].

In this review, we aim to illustrate the extent to which a new drug can affect the success, effectiveness, and safety of influenza therapy. The detailed presentation and comparison of all flu-specific antiviral drugs on the market will help answer these questions.

## 2. Influenza Virus

### 2.1. Structure of Influenza Virus

Influenza virus belongs to the Orthomyxoviridae family and has a spherical or filamentous shape, a viral envelope, and a segmented, negative-sense, single-stranded RNA genome [8,9,10]. There are two glycoproteins on the surface of the virus that control its ability to cause disease: hemagglutinin (HA) and neuraminidase (NA) [9,10,11,12,13]. These two proteins are antigens and define the specific influenza strain [14]. Another transmembrane protein is M2, which forms the ion channel. Under the glycoprotein envelope of the virus is a matrix made up of M1 protein, which shapes the virion and encloses its core [9,10,11,12,13]. Inside the matrix is the viral genome in the form of segmented RNA and non-structural nuclear export proteins (NEPs). Single-stranded viral RNA (vRNA) coated with nucleoprotein (NP) exists in a complex with RNA polymerase. RNA polymerase consists of three subunits (PB1, PB2, and PA) and is necessary for the transcription of viral genetic material during the replication of influenza virus [10].

### 2.2. Influenza Virus Life Cycle

After entering the respiratory tract, the influenza virus must travel through the thick layer of mucus covering the epithelium of the respiratory system. The main component of the mucus is oligosaccharides, which contain sialic acid. Viral NA hydrolyzes the α-glycosidic bond in the sialic acid molecules, relaxing the mucus and allowing the virions to bind to respiratory epithelial receptors [15]. The entry of the virus into the host cell by endocytosis is possible due to HA, which allows adhesion to the respiratory epithelial cell, as shown in Figure 1. This process involves the binding of HA to the sialic acid residues of surface receptors on the host cell. This leads to viral fusion and penetration of viral RNA into the cell interior [9,12,16]. After entering the host cell, the viral genetic material is released from the virion. This is possible due to the opening of M2 ion channels and acidification of the virus core. This acidic environment in the virion releases the viral RNA complex from the protein matrix into the host cell’s cytoplasm. Influenza vRNA is negative-sense RNA, which means that it must first be transcribed to positive-sense RNA before it can be used as a template for the production of vRNA. For this purpose, viral ribonucleoproteins (vRNPs) translocate to the interior of the host cell nucleus from the cytoplasm. Host mRNA and the RNA polymerase complex play key roles in the replication of influenza vRNA. The RNA polymerase complex consists of three PA subunits, PB1, and PB2, which are involved in transcription as follows. Due to the PA subunits, the cap of host mRNA is hydrolyzed and detached, and the primers for vRNA transcription are created. Next, the PB2 subunit binds the 5′ end of host mRNA, which allows the initiation of transcription of viral mRNA using the vRNA template. The PB1 subunit of the polymerase is responsible for the synthesis of mRNA strands [10,15,17]. The viral mRNA is exported to the cytoplasm of the cell, where translation of the viral proteins HA, NA, M1, and M2 takes place. At the same time, new copies of vRNA are transcribed in the host cell nucleus. The increased amount of synthesized M1 protein leads to the enhancement of nuclear vRNA export with the participation of NEPs, NP exporting proteins, and host proteins. In addition, by means of the COPI coat protein complex, HA, NA, and M2 proteins are transported to the apical part of the host cell membrane. M1 together with vRNA is transported to the cytoplasm. The M1 protein and vRNPs accumulate on the inner side of the membrane of the apical part of the host cell. Next, due to the interaction with the M1 protein, segments of vRNA are packed into virions [15]. A budding zone of the cell membrane is created as a result of the accumulation of proteins in this zone of the cell. The M2 protein is involved in the shape formation and detachment of the bubble. Newly replicated viruses are released due to NA, a tetrameric protein that has enzymatic activity and catalyzes the breaking of the alpha-ketosidic bond between sialic acid and adjacent sugar residues [12,15,18,19]. Without the participation of NA, only one round of replication would take place, significantly reducing, or even inhibiting, the infection and preventing the development of disease [14].

## 3. Drugs Used for Influenza Treatment and Prophylaxis

### 3.1. Amantadine

Amantadine was the first antiviral drug used in the treatment of influenza. Due to its mechanism of action, it can only be used against influenza A [20,21]. This drug inhibits viral replication by blocking the A/M2 proton channel specific to influenza A virus. However, naturally occurring point mutations in the transmembrane domain, which occur relatively quickly, have resulted in new amantadine-resistant strains of influenza A [22,23].

Amantadine has a dose-dependent effect and two active forms: amantadine hydrochloride for oral use and amantadine sulfate for oral and intravenous use [24]. The first is used as antiviral treatment. In general, amantadine is well tolerated, but due to the risk of impaired excretion in the case of renal failure, it is recommended to start treatment with the lowest doses and doses adequate for creatinine clearance [25]. The effectiveness of amantadine antiviral treatment is estimated to be a 50% reduction in the duration of symptoms if therapy is started in the first 48 h of infection with amantadine-sensitive influenza A virus [21,26]. Due to increased global resistance, amantadine has not been recommended for the treatment of influenza since 2006 [8,27,28]. However, it is still an area of interest and has other registered indications [29,30].

Currently, amantadine is mainly used in neurodegenerative diseases, such as Parkinson’s disease [31], therapy after traumatic brain injury [32,33], and multiple sclerosis [20,34]. Recently, due to the SARS-CoV-2 pandemic and the lack of specific treatment in this direction, attempts have been made to use amantadine in the fight against this deadly virus. It was noted that people with neurological diseases chronically treated with amantadine experienced asymptomatic SARS-CoV-2 infection [35]. A preliminary therapeutic effect of amantadine in patients with severe disease has also been demonstrated [36]. However, documented data remain insufficient to draw clear conclusions, and amantadine is not recommended in COVID-19 therapy.

### 3.2. Neuraminidase Inhibitors (NAIs)

#### 3.2.1. Group Presentation

NAIs are the largest group of drugs and are currently the most commonly prescribed and used drugs in the treatment of human influenza [37,38,39]. Their mechanism of action, consisting of the inhibition of viral replication in vitro and in vivo, was first demonstrated by von Itzstein et al. in 1993 based on the example of zanamivir [40]. Zanamivir entered clinical trials 1 year later [41] and was first approved by the Food and Drug Administration (FDA) for the treatment of influenza A and B infections on 26 July 1999 as a powder formulation for oral inhalation [1]. In 2006, it was approved for the prevention of influenza A and B [37]. Another NAI described by Kim et al. in 1997 was oseltamivir [42], which was first registered for medical use in 1999 in the US for oral use as Tamiflu^®^ (Roche) [11]. The NAIs currently available for general use include zanamivir, oseltamivir, peramivir, and laninamivir [11,18,43]. All of them are effective against most strains of influenza A and B [18] and, unlike amantadine, are associated with low toxicity and are significantly less likely to promote the development of drug resistance [14].

NAIs should be administered within the first 48 h of symptom onset. This is related to the peak of viral replication in the respiratory tract, which occurs between 24 and 72 h after infection with influenza virus and is a key period to inhibit this process [14]. Although the best clinical benefit is obtained after using any NAIs within the first 2 days after the onset of symptoms, some studies have demonstrated a good clinical response, even up to 5 days after symptom onset [44,45,46]. Numerous studies have shown that the use of NAIs shortens the length of treatment and hospitalization in seriously ill patients in intensive care units (ICUs) and reduces the risk of death [37,47,48,49,50,51,52]. Currently, for patients with a suspected or confirmed influenza infection, it is recommended to start treatment with drugs from this class as soon as possible [49]. NAIs are also available for prophylaxis, and their use is especially recommended in people suffering from high-risk flu during influenza season. However, such use, though important and necessary, is only a support for vaccination, which remains the main form of prevention against influenza [37,47,53].

Despite the significant efficiency of all NAIs, there is a constant need to improve and synthesize new variants due to the rapid, spontaneous, and uncontrolled variability in the influenza virus and the emerging drug resistance of some new strains [47,54].

#### 3.2.2. Mode of Action

NA is an exosialidase composed of a polypeptide chain containing 470 amino acid residues. The structure of the protein consists of the head, stalk, transmembrane domain, and cytoplasmic domain. On the tetrameric head are active sites that are essential for the hydrolysis process [55]. Each of the four protein monomers is anchored to a common point in the viral envelope. All four have a catalytic site with a linkage of sialic acid residues located in a deep, negatively charged pocket [11,16]. It is a calcium-binding domain made of oxygens from the rest of the main chain and the side chain. In addition, the active sites responsible for the catalytic function of the enzyme contain acid residues that are strictly conserved in influenza A and B viruses: Arg118, Arg152, Arg224, Arg292, Asp151, Glu276, Arg371, Tyr406, and the Asn146 glycosylation site in complex with zanamivir (Figure 2). The latter is distinguished by the presence of o-4 N-acetylgalactosamine sulfate. The hydrolytic activity of A consists of the formation of an oxocarbonate ion at the C2Neu5AC atom of the substrate. The introduction of the C2Neu5AC residue to the active site leads to strong ionic interactions between the substrate carboxylate and arginine guanidine groups, which then leads to a change in the conformation of Neu5Ac and cleavage of the glycosidic bond [55].

Numerous viral mutations are associated with amino acid changes in the NA structure, but the sequences that build the active site of the cleft remain highly conserved. This fact was used in the design of antiviral drugs by using the conserved active site as their target [16,56].

Analogs of 2,3-dehydro-2-deoxy-N-acetylneuraminic acid (DANA) with a C4-OH substitution and a 4-guanidino group mimic the transient state of the hydrolysis reaction. The interaction is the formation of a strong hydrogen bond between the carboxylic acid of the ligand and the residues Arg118, Arg292, and Arg371 in NA. Moreover, a specific hydrophobic contact occurs between the methyl group and two residues, Trp178 and Trp222 [55]. Due to this binding to the amino acid residues, NAIs block the active site of the enzyme via the most energetically advantageous interaction [9,11,14,16,37,57].

The first NAI that was developed was zanamivir. The structure of its molecule is based on DANA (as shown in Table 1) and a positively charged guanidino group, which binds to the active site of NA in a highly conservative manner, causing its inhibition [11]. Oseltamivir was also designed based on a modification of the DANA structure. However, unlike zanamivir, it has a cyclohexene ring, C4 amino group, and lipid side chain [11,12,14]. This allows oral administration of oseltamivir, which, as a phosphate prodrug, undergoes hepatic metabolism by carboxylesterases to its active form, oseltamivir carboxylate (OC) [12,16,58,59]. Peramivir’s structure and mechanism of action differ slightly from those of other NAIs. Based on DANA, it has a cyclopentane ring, a guanidino group, and a hydrophobic side chain [11,12]. As a result, it establishes multiple interactions with the NA catalytic site, which makes it active against some strains of influenza A and B viruses resistant to older drugs of this class [37]. Peramivir is administered as an intravenous infusion. The only long-acting NAI is laninamivir, the synthesis of which is based on zanamivir with a 7-OCH3 substitution. It is administered in a single dose by oral inhalation [11], and its main advantage is activity against some oseltamivir-resistant strains of influenza virus [37].

Regardless of the structure of the molecule, all NAIs mimic the sialic acid transition state by binding specifically to the active site of NA, leading to blockage of its enzymatic function [9,14]. As a consequence, viral replication stops at the stage of viral envelope formation, preventing the release of progeny virions from host cells and their subsequent spread [9,11,14,37,57].

#### 3.2.3. Zanamivir

##### Drug Presentation

Currently, zanamivir is approved for use under the two trade names Relenza^®^ (GlaxoSmithKline, GSK) and Dectova^®^ (GSK) [60,61,62,63]. Relenza is available in 70 countries around the world as an anti-influenza A and B treatment and preventative [56]. Dectova^®^ is a new preparation of zanamivir as a solution for infusion and authorized throughout the European Union [57].

##### Pharmacokinetics of Zanamivir for Oral Inhalation

Zanamivir is administered as a powder for oral inhalation (Relenza^®^, GSK). The main excipient is lactose monohydrate, which contains milk proteins [61,64]. Zanamivir is mixed with lactose in a proportion of 1:4, which is 5 mg of active substance per 20 mg of excipient [12,64]. Due to its large molecule size (>40 µm), lactose causes the main part (about 78%) of the drug to deposit in the mouth and throat, which is the first location of infection. The rest of the inhalation, ~13–15%, is deposited in the tracheobronchial tree and the lungs [12]. The drug concentration in the respiratory tract has been estimated to be 1000 times higher than the 50% inhibitory concentration (IC50) for NA [14]. Absolute bioavailability is very low following oral administration (1–5%). Following oral inhalation, bioavailability is 4% to 17%. The volume of distribution approximates that of extracellular water. Zanamivir plasma protein binding is limited and estimated to be <10%. Zanamivir is not metabolized; the serum half-life ranges from 2.5 to 5.1 h, and it is excreted unchanged in the urine by the kidney, with the excretion of a single dose completed within 24 h [65].

##### Treatment and Prophylaxis with Zanamivir for Oral Inhalation

The active component of Relenza^®^ is zanamivir; its pharmaceutical form is white to off-white powder contained in a disc blister, pre-dispensed with 5 mg of zanamivir per dose and administered using the Diskhaler, a specially designed breath-activated plastic device [61,64]. Each inhalation that is deposited in the mouth delivers 4.0 mg of zanamivir to the respiratory tract.

Zanamivir for inhalation is licensed for treatment of influenza A and B in patients with typical symptoms during periods in which influenza virus is circulating in the community. The drug is authorized for treatment in adults and children (≥5 years) in most countries, including the European Union and Australia, and in patients aged 7 years and older in the United States and Canada [61,64,66]. The recommended dosage for treatment of influenza is two inhalations of 5 mg each twice daily (approximately 12 h apart) for 5 days. Treatment should be started as soon as possible, no later than 36 h after the onset of symptoms in children and 48 h after the onset of symptoms in adults. Two doses should be taken on the first day with at least 2 h between doses [61,64].

For seasonal prophylaxis against influenza, zanamivir for oral inhalation is approved in most countries for adults and children over 5 years of age and in Canada for adults and children aged 7 years and older. The recommended dose is 10 mg of zanamivir once daily for 28 days [61,64]. It is also approved for post-exposure prophylaxis against influenza A and B in an individual exposed to clinically symptomatic household members. In this case, it is recommended to use the drug up to 36 h after contact with the patient at a dose of 10 mg once daily for 10 days (approximately 24 h apart) [61,64].

The drug is also approved for prophylaxis in the event of an epidemic or pandemic of influenza A and B, such as when there is a mismatch between the strains contained in the vaccines and those circulating in the general population. However, zanamivir is not a substitute for influenza immunization, and its prophylactic use should always be determined on a case-by-case basis.

A contraindication to the use of zanamivir for oral inhalation is an allergy to milk protein. In addition, patients with glucose-galactose malabsorption, hereditary galactose intolerance, and Lapp lactase deficiency should not use this medicine. There have been reports of unsuccessful attempts to administer Relenza^®^ by nebulization or mechanical ventilation due to the drug’s lactose content. Consequently, it obstructs the action of the equipment [61,64]. Due to very rare reports of bronchospasm and declining respiratory function after drug administration, its usage in patients with asthma and chronic obstructive pulmonary disease (COPD) requires great care. In these special cases, the patient should be advised of the risk of bronchospasm and the need for a fast-acting bronchodilator. Due to limited data, it is not possible to definitively determine the efficacy and safety of zanamivir for oral inhalation in patients with asthma, chronic respiratory disease, immune deficiencies, or unstable chronic illnesses; in pregnant and lactating women; in elderly patients over 65 years of age; and in the prevention of influenza in nursing homes [61,64,67].

Zanamivir for oral inhalation has been associated with the potential for adverse effects (AEs) and complications. Some of the more common reactions that can occur with the use of zanamivir inhalation powder include skin reactions, such as rash. Studies have shown that such reactions may occur in 1% to 10% of patients. Uncommon AEs that have been reported include disordered respiratory function, bronchospasm, throat tightness or constriction, vasovagal-like reactions, and allergic-type reactions, including oropharyngeal edema and urticaria. The frequency of this type of reaction is estimated to be 0.1% to 1%. Rare AEs have been observed in less than 0.01% of patients and include anaphylactic reaction, facial edema, toxic epidermal necrolysis, erythema multiforme, and Stevens–Johnson syndrome [64] (Table 2). In addition, there are post-marketing reports of incidents of psychiatric and nervous system disorders in influenza patients treated with zanamivir for oral inhalation. Seizures, delirium, hallucination, abnormal behavior, and depressed level of consciousness have been described [62,64].

##### Pharmacokinetics of Intravenous Zanamivir

After intravenous administration of Dectova^®^, zanamivir binds plasma proteins to a very low degree (<10%), and the central volume of distribution is approximately 16 L, which approximates the volume of extracellular water. The drug does not accumulate in serum, and proportionality has been demonstrated between its maximum concentration (Cmax) and the area under the curve (AUC). Zanamivir is not significantly metabolized and is eliminated unchanged in the urine by glomerular filtration, with a half-life of approximately 2–3 h in adults with normal renal function after repeated intravenous doses of 600 mg twice daily [77,78,79,80].

##### Treatment with Intravenous Zanamivir

Dectova^®^ is a zanamivir solution for infusion designed to treat seriously ill, hospitalized patients with life-threatening infections with influenza A or B virus [43,65,81]. This medicine was approved by the European Medicines Agency (EMA) in 2019 and has the status of a drug under additional monitoring, which means that Dectova^®^ is subject to constant reporting on its effectiveness and safety, as well as the analysis of its benefit–risk profile in everyday medical practice [63].

Intravenous zanamivir is registered for treatment in patients with known or suspected influenza virus infection resistant to other antiviral drugs and in patients whose medical condition does not allow the use of medications suitable for oral administration or inhalation, such as patients with sepsis, intestinal obstruction, or malabsorption [43,65,78,82,83,84,85]. Zanamivir is effective in inhibiting the replication of most oseltamivir-resistant influenza viruses, including strains with one of the most common H275Y mutations in N1 viruses [66,78,86].

Dectova^®^ is available on the market in 1% solutions of zanamivir (as hydrate); each vial of clear, colorless intravenous solution contains 200 mg of zanamivir in 20 mL dilution. An excipient with known effects is sodium (70.8 mg per vial) [43,65]. Intravenous zanamivir is licensed for adults, adolescents, and children over 6 months old. The recommended dose is 600 mg twice daily for 5–10 days, and treatment should be started in the first 6 days after the onset of influenza symptoms. However, weight-dependent treatment is recommended in infants, children, and adolescents weighing ≥ 50 kg as follows: children up to 6 years old, 14 mg per kg body weight twice daily for 5–10 days; children ≥6 years to <18 years old, 12 mg per kg body weight twice daily for 5–10 days. In patients with renal impairment, the dose should be individually adjusted according to the degree of renal failure and creatinine clearance [43,65,78].

Phase 2 and 3 clinical trials reported some adverse effects related to the drug (Table 2). The most common were gastrointestinal events, including diarrhea, rash, hepatic reactions such as hepatocellular injury, and increased levels of transaminases (ALT and AST). In children, neutropenia and renal failure were reported. Studies have shown that the frequency of common AEs during treatment with intravenous zanamivir is 1% to 10%. Uncommon AEs include urticaria and increased alkaline phosphatase, affecting up to 1 in 100 patients. The possibility of anaphylactic reaction, facial and oropharyngeal edema, neuropsychiatric reactions (such as abnormal behavior, delirium, and depressed level of consciousness), renal impairment, paralytic ileus, and cardiovascular events, including hypotension, have also been noted, but due to limited data, the frequency of occurrence could not be determined [43,65,66].

#### 3.2.4. Oseltamivir

##### Drug Presentation

Oseltamivir is an orally administered antiviral drug approved under the original trade name Tamiflu^®^ (Roche). It is the most commonly prescribed and utilized NAI for the treatment of influenza A and B infections [49,87,88].

##### Pharmacokinetics

Oseltamivir is administered orally as the prodrug oseltamivir phosphate salt and rapidly absorbed by the gastrointestinal tract. Hepatic esterases extensively hydrolyze it into the active form of oseltamivir carboxylate (OC) [9,16,18,88,89]. At least 80% of an oral dose reaches the systemic circulation as the active metabolite [9,16,89]. Absolute bioavailability is proportional to the dose and is similar in severely ill patients and those with mild symptoms [90,91]. It is unaffected by co-administration with food [9,12,14]. The mean volume of distribution is approximately 25.6 L in humans, a volume that is roughly equivalent to the extracellular body fluid. The binding of the prodrug is 42%, and only approximately 3% of the active form of OC binds to human plasma protein [67]. After oral administration at a dose of 150 mg, OC has an AUC of 6834 μg/L/h, Tmax of 2.88 h, and Cmax of 2091 μg/L [9]. The main route of oseltamivir elimination is generally via the kidneys (>99%), and the drug is excreted in the urine as OC [9,16] and requires dose reduction in patients with renal failure [49,58,88]. Plasma concentrations of oseltamivir decline with a half-life of 6.7–8.2 h after oral administration. Renal clearance of the drug, 18.8 L/h, exceeds the glomerular filtration rate of 6.67 L/h, indicating that tubular secretion occurs in addition to glomerular filtration [9,67].

##### Treatment and Prophylaxis with Oral Oseltamivir

Tamiflu^®^ is available in the pharmaceutical form as 30 mg, 45 mg, or 75 mg hard capsules and 6 mg/mL powder for oral suspension [67,68]. The EMA approved oseltamivir for treatment in adults, adolescents, and children, including full-term neonates, who show symptoms typical of influenza infection during periods of increased viral activity in the environment [49,67,68]. In the United States, the drug is approved for the treatment of uncomplicated influenza A and B in patients older than 2 weeks of age [58,59,68]. Similar to zanamivir, treatment with oseltamivir should be started 48 h after the onset of symptoms [67,68,69]. However, in special cases in hospitalized, seriously ill patients and in those at high risk of complications, it is recommended to initiate treatment with oseltamivir regardless of the time of onset. Such recommendations are based on studies that have shown a beneficial therapeutic effect of oseltamivir in hospitalized patients with influenza who started treatment 4 and 5 days after the onset of symptoms [59]. The registration of oseltamivir is also for prophylaxis. The EMA and FDA authorized the use of the drug in people older than 1 year of age in the case of contact with a person clinically diagnosed with influenza [67,68]. In a case of a global spread of the influenza virus, such as during a pandemic, when the vaccine does not provide the required protection, the EMA allows Tamiflu^®^ prophylaxis for the whole population, including full-term infants [67]. However, regardless of the circumstances, special care should be taken to avoid using oseltamivir for prophylaxis in individuals who have been treated with this drug in the past due to the risk of new viral mutations [37]. In addition, the WHO indicates oseltamivir as the drug of choice for the treatment and prevention of influenza caused by virulent influenza strains, including influenza A (H1N1) [92].

Acute, subacute, and chronic toxicity studies have demonstrated a high margin of safety for oseltamivir and no oncogenic and mutagenic effects [93,94]. Overall, the drug is well tolerated, and the most common adverse effects are headache and digestive system disorders [12,59,67,68,69]. Nausea and vomiting have been observed most frequently on the first day of treatment after the first dose, usually lasting for up to 2 days and resolving spontaneously [9,14]. The frequency of such symptoms increases with increasing dose and is probably associated with local irritation of the gastric mucosa [9,12]. Taking oseltamivir with food does not affect its bioavailability and is recommended to reduce negative gastrointestinal symptoms [9,12,14,16,70]. Other reported side effects, though much less likely to occur, are very serious and include anaphylactic reactions, toxic epidermal necrolysis, cardiac arrhythmia, hepatic failure (including hepatitis), gastrointestinal bleeding [59,67,68,69], and also other equally serious but usually not life-threatening effects (e.g., dermatitis, rash, visual disturbances, and evaluated liver enzymes) [67,69]. Notably, neuropsychiatric adverse events (NPAEs) have been reported during oseltamivir treatment in patients with influenza [37,53,58,59,69,95]. Such events have been reported in Japan, where the drug is widespread and used for the early treatment of influenza and the prevention of epidemic influenza. Due to numerous reports of self-injury, abnormal behavior, and some fatalities in adolescents, Japan’s Ministry of Health, Labor and Welfare (MHLW) issued a warning against the use of oseltamivir in March 2007. In response to this situation, the FDA analyzed data reports from Japan and the US regarding patients who experienced NPAEs from August 2005 to June 2006. Due to the lack of strong evidence, a strict relationship between the use of the drug and the occurrence of NPAEs cannot be concluded [96]. However, patients with influenza, especially children and adolescents treated with oseltamivir, should be closely monitored [96], and the expected symptoms include abnormal behavior, delirium, hallucination, self-injury, agitation, anxiety, delusions, and confusion [67,69].

To treat adults, oseltamivir is used at a dose of 75 mg twice a day for 5 days [16,37,67,68,69,88]. The exceptions are patients on immunosuppression, for whom it is recommended to extend the treatment to 10 days [67,69]. According to data from pharmacological studies showing differences between children and adults, in children and adolescents younger than 13 years, the dose should be adjusted for weight and age group [58]. Furthermore, among patients with renal failure, the dose of oseltamivir should correlate with creatinine clearance, the degree of kidney dysfunction, and body weight [49,58,67,68,69]. Finally, obese adults, including those with a BMI > 40 kg/m^2^, do not require an increased dose [49].

##### Oseltamivir as an Over-the-Counter Drug

As influenza is a serious health, economic, and epidemiological problem in the US, attempts were made to register oseltamivir as an over-the-counter drug in 2019. There would be many advantages of such a solution. General access to the drug could shorten the time from the first symptoms of the disease to treatment application, which seems to be a key factor in the effectiveness of therapy. This could significantly relieve the burden on health services and reduce the number of severely ill patients requiring hospitalization. On the other hand, there is concern about drug abuse that could lead to drug resistance. There are also some risks associated with diminished public interest in vaccine prophylaxis. Thus, balancing the benefits and losses is extremely important in the final decision to register oseltamivir as an over-the-counter medicine [40]. At the moment, such regulation has not entered into force, but this may change soon [97].

Several generics for Tamiflu^®^ are currently available on the European market, including EMA-approved Ebilfumin. All of them require a prescription [95,98].

#### 3.2.5. Peramivir

##### Drug Presentation

Peramivir was approved for use in Japan in 2010 under the trade name Rapiacta^®^ (BioCryst Pharmaceuticals (BCP)) and the same year in South Korea as PeramiFlu^®^ (BCP). Subsequently, the drug was approved in the United States in 2014 as Rapivab^®^ (BCP) and in the EU in 2018 under the trade name Alpivab^®^ (BCP) [71]. However, Alpivab^®^ has been withdrawn from the EU market at the request of the manufacturer for commercial reasons [99].

##### Pharmacokinetics

The pharmacokinetic parameters following intravenous administration of peramivir at a dose of 800 mg or 400 mg twice daily exhibit a linear relationship between dose and exposure parameters (Cmax and AUC) [100]. Peramivir binds plasma proteins at a rate of <30%, and the central volume of distribution is approximately 12.56 L. Peramivir is not significantly metabolized in humans. The major route of elimination of peramivir is via the kidney. In patients with normal renal function, peramivir intravenously administered as a single 600 mg dose is eliminated with a half-life of approximately 20 h. Renal clearance of unchanged peramivir accounts for approximately 90% of total clearance [101].

##### Treatment with Intravenous Peramivir

Rapivab^®^ is currently the only NAI approved by the FDA for the treatment of acute uncomplicated influenza and is provided in solution for intravenous infusion [71,97]. The uniqueness of this drug lies in the use of one parenteral dose [71,72,73]. The potential benefits of a single dose and intravenous administration make it possible to use this drug in patients with swallowing problems and in people with nausea and vomiting. In addition, the administration of only one dose significantly simplifies the treatment regimen and excludes the possibility of the patient skipping a dose of the drug [72].

Intravenous peramivir is approved for the treatment of acute uncomplicated influenza in adults and children over 2 years of age [94]. The recommended period from the onset of symptoms to initiation of treatment is 48 h, the same as for other NAIs [73,94], but there are some reports of benefits from later peramivir use [71]. The drug is available as a colorless solution of 1% peramivir (200 mg in 20 mL). The intravenous peramivir dose is 600 mg once in adults and adolescents and 12 mg/kg of body weight in children from 2 to 12 years of age [94]. Patients with renal impairment require adjustment of the dose to the level of creatinine clearance. Depending on the degree of glomerular filtration disorder, the dose does not change or amounts to one-third or one-sixth the standard dose [72,73].

Studies are being conducted on the effectiveness of peramivir treatment in seriously ill patients. It seems necessary to modify the dose and increase the number of doses in this group of patients. However, it requires further research [72,73].

Peramivir is generally well tolerated, and the most common adverse effects in adults are neutropenia and diarrhea (Table 2). Other AEs, though less common in this patient group, are constipation, insomnia, increased AST, and hypertension [71,72,73]. Among pediatric patients, the most common AEs following the use of intravenous peramivir are digestive system disorders, such as diarrhea (5–33%), nausea (2.5%), and vomiting (0.5–2%) [74]. Other adverse reactions that are less common but specific for patients up to the age of 18 years include injection site rash, fever, proteinuria, and tympanic membrane erythema [72,73]. There have also been reports of AEs from post-marketing experience, which are recorded voluntarily, and therefore, it is difficult to determine their frequency and relationship with drug exposure. These adverse effects include anaphylactic reactions, severe dermatological reactions such as Stevens–Johnson syndrome and exfoliative dermatitis, and NPAE-type reactions [72,73]. The potential influence of peramivir on the efficacy of live attenuated influenza vaccines should also be mentioned. It is recommended to avoid vaccinations for 2 days after administration of the drug. Furthermore, peramivir given within 2 weeks of influenza vaccination may adversely affect the immunogenicity of the vaccine [73].

#### 3.2.6. Laninamivir

##### Drug Presentation

Due to the serious influenza burden in Asia, great effort is being made in this part of the world to develop effective methods of reducing influenza virus. The result of this work is laninamivir octanoate (LO), a long-acting NAI, discovered and developed by the Japanese pharmaceutical company Daiichi Sankyo. Japan is the first country in the world to approve laninamivir for the treatment of acute uncomplicated influenza A and B. The drug was approved in July 2010 and was launched on the market in October 2010 under the trade name Inavir^®^ (Daiichi Sankyo) [75,102]. Since then, Inavir^®^ has been widely used every year to treat seasonal flu infections in Japan, and since 2013, it has been used for prophylaxis against influenza. However, it was never registered and is not available on the European and American markets [102,103,104].

##### Pharmacokinetics

LO is an octanoyl prodrug of laninamivir administered as a powder for oral inhalation. After administration to the lungs, the prodrug is hydrolyzed to its active form, and its concentration remains high for a long time [103]. After a single inhaled dose of 40 mg of LO, the plasma half-life was estimated to be 64.7–74.4 h in healthy volunteers and 53.2–57.0 h in patients with renal insufficiency. The drug concentration in the respiratory tract has exceeded the IC50 for influenza virus NA even up to 240 h after inhalation. The mean plasma Cmax and AUC increase proportionally between doses. The drug concentration in the respiratory tract has been estimated to be 10,000 times higher than the concentration in plasma. The cumulative amounts of urinary LO and laninamivir excreted over 144 h after LO inhalation account for >15% of the human dose. Preclinical animal studies in rats demonstrated that fecal excretion was approximately 36% following intravenous administration of LO, and the absolute oral bioavailability in rats was minimal (unpublished data). Binding to LO and laninamivir proteins was measured in an in vitro experiment using human plasma and found to be 67% and <0.1%, respectively (unpublished data) [105,106]. The detailed pharmacokinetics of laninamivir have not yet been fully described. Further research is needed.

##### Treatment and Prophylaxis with Inhaled Laninamivir

Inavir^®^ is a prodrug laninamivir octoate available as an oral inhalation powder administered by a disposable dry powder inhaler [104,107]. Therapy with laninamivir is based on a single dose of the drug, which is an advantage over zanamivir and oseltamivir therapy. The recommended dose for the treatment of influenza A and B is 20 mg laninamivir in children < 10 years old and 40 mg in patients ≥ 10 years old [102,103,104]. However, studies presented by Murasaka et al. (2017) showed that patients with low peak inspiratory flow are unable to inhale the recommended dose of the drug. Therefore, the drug information sheet suggests that this group of patients requires more inhalations in order to deliver the standard dose to the lungs [107]. For prophylaxis against the influenza virus, the recommended dose is 20 mg of laninamivir daily for 2 days for children over 10 years of age, adolescents, and adults [102].

Laninamivir is generally well tolerated in both pediatric and adult patients. No fatalities and no serious AEs have been reported. The main symptoms related to drug administration are cough (4.72%), diarrhea (3.77%), headache (3.30%), and gastritis (0.47%), according to clinical trials [76]. The incidence of adverse reactions assessed during post-approval drug safety surveillance was 1.41% and mainly related to gastrointestinal disorders, such as nausea, vomiting, and diarrhea, and abnormal behavior (0.45%), as well as nervous system disorders such as dizziness (0.17%) [75] (Table 2).

#### 3.2.7. Comparison of the Effectiveness of NAIs

The oldest and best known and studied NAIs are zanamivir and oseltamivir. There have been many studies on their effectiveness and safety conducted over 20 years. Over time, the effectiveness of treatment with these two drugs has become a reference point for the evaluation of subsequent NAIs launched on the market, such as peramivir and laninamivir [37,71,72,86,108,109,110].

Based on two randomized, double-blind, and placebo-controlled trials, Hayden et al. showed the efficiency of zanamivir. Zanamivir for oral inhalation demonstrated the highest clinical response in patients who started treatment up to 30 h after the onset of symptoms. The mean time to relief of the main symptoms was shortened by 1 day compared to placebo, and in the group of patients with fever who started treatment early (up to 30 h), the time to relief of symptoms was reduced by 3 days. It was noted that treatment had better results in patients with fever [41]. On the other hand, the results of the meta-analysis by Heneghan et al., who analyzed 26 studies with zanamivir (Relenza^®^), showed that the mean duration of treatment with zanamivir for oral inhalation in adults reduced the time to relief of symptoms by 14.4 h, which is equivalent to 0.6 days [53]. Another study, the double-blind MIST, included children over 12 years of age and adults with influenza symptoms and patients with influenza A and B confirmed by a laboratory test. Study participants were randomized to receive 10 mg of zanamivir for oral inhalation twice daily for 5 days or placebo. The median time to relief of symptoms in patients without fever who were treated with zanamivir was not significantly different from the placebo group, in contrast to febrile patients, who recovered 2 days faster than the placebo group. A significant difference was also observed in the group of high-risk patients, in whom relief of symptoms occurred 2.5 days earlier than in the placebo group on average. Treatment with zanamivir has been shown to be effective for both influenza A and B, and the importance of starting treatment within 36 h of the onset of symptoms has been emphasized. It was also noted that high-risk patients treated with zanamivir had fewer complications and required fewer antibiotic prescriptions than the placebo group [111]. Another study by Walker et al. drew attention to the beneficial effect of zanamivir for oral inhalation in reducing influenza complications. Early initiation of this drug has been shown to significantly reduce the risk of middle-ear disorders and middle-ear pressure disorders, which are common complications of influenza [112].

The prophylactic effects of zanamivir have been investigated in several large randomized, double-blind, placebo-controlled trials. All of them were prophylactic, with zanamivir administered at a dose of 10 mg once daily [113,114,115]. The efficacy of zanamivir in preventing seasonal influenza in healthy adults has been estimated at 67% for persons with laboratory-confirmed clinical influenza and 84% for persons with laboratory-confirmed illnesses with fever during seasonal prophylaxis [113]. Post-exposure prophylaxis, as assessed by Hayden et al., was determined to be 79% effective in protecting against influenza A and B [114]. Similar results were obtained by Monto et al., who estimated a protective efficacy of 82% against influenza A and B (78% and 85%, respectively) [115].

Phase 3 clinical trials conducted by Roche on the efficacy and safety of oseltamivir showed that 75 mg of Tamiflu^®^ twice daily for 5 days in the treatment of flu patients with fever shortened the median time to all symptom relief by 1.3 days compared to placebo. The study group consisted of adults who received the first dose of the drug within 40 h of symptom onset. In pediatric patients aged 1–12 years, this time was reduced by 1.5 days, provided that treatment was started in the first 48 h [67]. Subsequent studies showed that the treatment effect of oseltamivir resulted in a 38% reduction in disease severity and a 30% reduction in the median duration of illness onset versus the placebo group. Furthermore, recovery to normal activity was 2–3 days faster in patients treated with oseltamivir versus placebo [116]. A multicenter study from Japan revealed that treatment with oseltamivir started in the first 12 h after the onset of fever shortened the course of the disease by more than 3 days compared to treatment started after 48 h [14]. Whitley et al. showed that treatment with oseltamivir started 48 h after the onset of symptoms in a group of pediatric patients aged 1–12 years with influenza diagnosed based on clinical symptoms reduced disease duration by 36 h compared to placebo [117]. In this study, treatment with oseltamivir also reduced fever duration, cough, and coryza, with a 44% reduction in the risk of developing otitis media [117].

In another study by Hayden et al., treatment with oseltamivir shortened, though not significantly, the time to alleviation of symptoms by 16.8 h in the adult group and by 29 h in children with uncomplicated flu, but it had no effect in children with asthma [53].

Many factors, including mortality, risk of hospitalization, duration of hospitalization, and risk of complications in the form of pneumonia, have been considered and analyzed in the context of treatment with oseltamivir, but a clear answer remains to be found [12,14,37,53,58]. On the other hand, the effectiveness of seasonal and post-exposure prophylaxis in neonates, children, adults, and the elderly using oseltamivir is indisputable and confirmed by many studies to range from 70 to 90% [12,14,53,58,118,119,120].

Two randomized controlled trials of the combination of zanamivir and oseltamivir were conducted to improve the effectiveness of NAI treatment. Both provided similar results, with no additional benefits of these treatments. Interestingly, oseltamivir monotherapy turned out to be more effective than the combination therapy [121,122].

Zanamivir administered intravenously to hospitalized adult patients with severe influenza, including those requiring intensive care, reduced the number of patients in the ICU by 40% in the largest randomized double-dummy phase 3 trial [84]. Patients were divided into three groups depending on the type of therapy used: group 1, 600 mg of intravenous zanamivir twice daily; group 2, 300 mg of intravenous zanamivir twice daily; and group 3, 75 mg of oral oseltamivir twice daily for 5 days, with the option of extending the treatment for another 5 days if necessary. The median time to improvement was 5.14 days in group 1, 5.87 days in group 2, and 5.63 days in group 3. These differences were not significant, and treatment with intravenous zanamivir at both doses was considered as effective as treatment with oseltamivir. The overall mortality was 7%; it was similar in all groups and consistent with previous reports on NAI treatment in hospitalized patients [84]. A phase 2 open-label, multicenter, single-arm study of the efficacy of intravenous zanamivir in hospitalized children reported a 92% positive clinical response to the assumed endpoints and a 7% mortality rate [86].

Peramivir was first evaluated for safety and efficacy in the double-blind, placebo-controlled, phase 2 trial conducted by Kohno et al. Adult patients with uncomplicated seasonal influenza virus infection were divided into three groups to receive a single intravenous dose of peramivir (600 mg or 300 mg) or a placebo [123]. The study showed that the time to relief of symptoms was significantly shortened in patients treated with peramivir, regardless of the influenza subtype [123]. Subsequent large phase 2 and 3 studies comparing the efficacy of a single dose of intravenous peramivir versus oral oseltamivir (75 mg 2xd) showed a comparable effect on time to clinical stability and symptom duration in both seasonal uncomplicated influenza patients and hospitalized patients [124,125]. Moreover, Koho et al. showed that treatment with a single dose of peramivir shortens the duration of fever by 3 h compared to oseltamivir treatment (<34 h vs. <37 h, respectively) and may be an equivalent alternative to treatment with 5 days of oseltamivir [124]. Two retrospective studies demonstrated the superiority of peramivir over other NAIs in terms of fever duration in pediatric patients [126,127]. Both studies concerned the treatment analysis of all four NAIs (zanamivir, oseltamivir, peramivir, and laninamivir). The first study showed that, in children infected with influenza A/H3N2, the median duration of fever after treatment with peramivir was significantly (3.3 times) shorter than after treatment with oseltamivir [126]. The second study in children 5–18 years of age showed that, in children with influenza A treated with peramivir, the median duration of fever was 1 day shorter than after treatment with zanamivir and 2 days shorter in children with influenza B treated with peramivir compared to treatment with laninamivir [127].

A multinational phase 3 study compared the efficacy of laninamivir administered as a single inhaled dose at 20 mg or 40 mg versus oseltamivir phosphate at 75 mg 2 times per day for 5 days [128]. This study found that the effects of treatment with laninamivir and oseltamivir are comparable in regard to reducing the duration of symptoms and shedding the virus. Laninamivir at 40 mg demonstrated a better effect than 20 mg and was effective in the treatment of infections (especially of children) caused by oseltamivir-resistant influenza A/H1N1 (H274Y) [128]. In another large observational study, Mawatori et al. compared the efficacy of treatment with four NAIs in patients with influenza A/H1N1, A/H3N2, and B. The results showed that the laninamivir-treated group had a significantly longer duration of fever compared to patients treated with oseltamivir for both types of infections (influenza A and B) [129]. The results of this study are consistent with previous research [109,127,130]. In contrast, the use of laninamivir for the post-exposure prevention of influenza has significant benefits. A double-blind, randomized, placebo-controlled study conducted by Kashiwagi et al. showed a 62.8% reduction in the incidence rate compared to placebo [131].

### 3.3. Cap-Dependent Endonuclease Inhibitors—New Group of Anti-Influenza Drugs

#### 3.3.1. Baloxavir Marboxil

After nearly 20 years, a new class of anti-influenza drug was approved in 2018, first in Japan and then in the United States. Baloxavir marboxil is a cap-dependent endonuclease inhibitor (CENI), an antiviral drug against influenza A and B viruses, including oseltamivir-resistant strains [17,132]. This drug was registered in the European Union market in July 2021 under the trade name Xofluza^®^ (Roche), with the indication for treatment of acute uncomplicated influenza and post-exposure prophylaxis [133]. Recently, in the United States, the indications for the use of Xofluza^®^ have been extended to treating patients with a high risk of developing influenza-related complications [132].

#### 3.3.2. Mode of Action

Baloxavir marboxil is a small-molecule prodrug that, after oral administration, is hydrolyzed in the intestinal epithelial cells, blood, and liver to the active form baloxavir acid [7]. Active baloxavir inhibits influenza virus replication by selective binding of the RNA-dependent influenza virus RNA polymerase complex to the PA protein [15,17]. The transcription of vRNA is carried out in the nuclei of host cells by a specific polymerase. This enzyme is composed of three subunits: PB1, PB2, and PA [19]. Baloxaviric acid inhibits viral mRNA formation by specific binding of the PA protein, which is responsible for capturing the host cell mRNA cap and a key step in initiating the process of vRNA transcription [15,17,134].

The prodrug baloxavir marboxil has a phenolic hydroxyl group to enhance oral absorption. The drug was designed based on the Dolutegravir DTG molecule, a drug used in the treatment of HIV infections. When creating the baloxavir molecule, a DTG molecule chelating metals in the active site of viral integrase was used as a chemical scaffold. Limited information is available on the mechanism of baloxavir binding to PA and on potential determinants of reduced susceptibility [135].

Structurally, the baloxavir molecule resembles butterfly wings, where one of the wings contains a chelating frontal metal, the oxazine-pyridotriazinedione polar group, and the other wing contains a lipophilic difluoro-dihydrodibenzothiepin tail group that connects to the active site through van der Waals interactions. At the endonuclease active site, the head group binds to two divalent cations, and each metal ion has octahedral coordination through interactions with a protein. The first active site binds preferentially to manganese, and acid residues His41, Glu119, and Asp108 are present, whereas the second active site partially binds manganese and magnesium and has the acid residues Asp108 and Glu80. In influenza A, baloxavir forms van der Waals interactions with Ala20 and Tyr24 from the C-terminal end of the α-2 helix and Lys34, Ala37, and Ile38 from the α-3 helix, as shown in Figure 3.

In influenza B, however, similar interactions occur with other residues: Thr20, Phe24, Met34, Asn37, and Ile38. Both types of van der Waals interactions take place through CG1 and CD1 atoms. In addition, a hydrogen bond is formed with the carbonyl oxides of the main Met34 and Leu35 chain. Due to these bonds and interactions with the PA protein molecule, the endonuclease is inhibited [136].

#### 3.3.3. Pharmacokinetics of Baloxavir

Baloxavir marboxil is administered orally as a prodrug that is hydrolyzed to baloxaviric acid in the gastrointestinal tract. Arylacetamine deacetylase (AADAC) is responsible for metabolizing the drug in the lumen of the gastrointestinal tract, intestinal epithelium, and liver. Maximum plasma concentrations of baloxaviric acid are reached 3.5–4 h after administration. Over the dose range of 6–80 mg, baloxaviric acid exhibits linear pharmacokinetics and is eliminated in the liver by glucuronidation and, to a lesser extent, cytochrome P450 3A4 oxidation. The mean terminal elimination half-life is 80 h. More than 80% of the drug is excreted in the feces [137,138].

#### 3.3.4. Treatment and Prophylaxis with Oral Baloxavir

The active component of Xofluza^®^ (Roche) is baloxavir marboxil. Its pharmaceutical form is white to pale yellow, film-coated, oval-shaped tablets for oral use. In the countries of the European Union, the indication for the use of the drug is the treatment of acute uncomplicated influenza and post-exposure prophylaxis, whereas in the United States, Xofluza^®^ has not been approved for prevention and is indicated for the treatment of acute uncomplicated influenza in healthy patients and patients at high risk of developing influenza-related complications. The drug is intended for use in adolescents and adults 12 years of age and older. According to medical guideline recommendations, oral baloxavir should be administered within 48 h of illness onset or up to 48 h after close contact with a person infected with the influenza virus [132,139]. Due to the long half-life of baloxavir, >79 h, a single-dose regimen is recommended. The dosage is based on body weight: one dose of 40 mg for a patient weighing 40–79 kg and one dose of 80 mg for a patient weighing ≥ 80 kg (Table 3) [61,64,65,66,73,102,104,109,139]. The drug can be taken with food, except those which contain high levels of calcium. It is also not recommended to combine baloxavir treatment with the administration of a live attenuated influenza vaccine [132,139].

#### 3.3.5. Safety of Baloxavir

Safety data for baloxavir marboxil come from two randomized, double-blind, phase 3 clinical studies, CAPSTONE-1 and CAPSTONE-2, with the latter enriched with a group of patients at high risk of influenza complications. Analysis of data from these studies revealed a significantly lower incidence of AEs reported in the baloxavir group compared to the oseltamivir group. The most commonly reported AEs included diarrhea, nausea, bronchitis, sinusitis, and headache. They occurred in at least 1 out of 100 participants [7]. Other reports from post-marketing experience with baloxavir have shown the possibility of serious adverse reactions, such as anaphylactic shock; anaphylactic reactions; swelling of the face, eyelids, and tongue; erythema multiforme; rash; colitis; bloody diarrhea; and psychiatric events, such as abnormal behavior, delirium, and hallucination. The frequency and correlation between AEs and baloxavir cannot be determined due to the voluntary nature of these reports from unknown population sizes [7,132].

Further research is currently being carried out on the safety and effectiveness of baloxavir. The results from one such study, the phase 3 clinical trial MiniSTONE 2, enrolling pediatric participants with influenza-like symptoms aged 1–12 years, are similar to those reported in adults. The most common AEs were gastrointestinal disorders (diarrhea and vomiting), and the frequency of all reported effects was 46.1% for treatment with baloxavir and 53.4% for treatment with oseltamivir [140]. However, the drug is not registered for the treatment of patients in this age group and requires further research before a possible extension of the registration [132,133].

#### 3.3.6. Effectiveness of Baloxavir

The first study of baloxavir antiviral efficacy was a phase 2 study examining the dose–response and antiviral activity of the drug. The results of this study were promising, as they showed a reduction in time to symptom relief in all baloxavir groups compared to placebo and a significantly greater reduction in virus titer within 24 h after baloxavir treatment compared to placebo; the data allowed the identification of the best baloxavir dose (40 mg and 80 mg weight-dependent) for further phase 3 studies [141].

Another study was CAPSTONE-1, a multicenter, double-blind, randomized study conducted to evaluate the efficacy of a single dose of baloxavir marboxil. The study participants were 1436 healthy patients with influenza aged 12–64 years, separated into adolescents (12–19 years) and adults (20–64 years). Adolescents were randomized into two groups, baloxavir and placebo, whereas adults were separated into three groups: baloxavir, oseltamivir, and placebo. Depending on the group, the following treatment was applied: single-dose baloxavir at 40 mg or 80 mg, 5-day oseltamivir at 75 mg twice daily, or placebo. Treatment was started within the first 48 h after the onset of flu symptoms. The results revealed the superiority of baloxavir over placebo on par with oseltamivir. Compared to placebo, both baloxavir and oseltamivir showed a significant reduction in time to the alleviation of symptoms (placebo 80.2 h, baloxavir 53.7 h, and oseltamivir 53.8 h; Table 4). Baloxavir also resulted in a significant reduction in the median time to fever resolution; the effect was achieved 17.5 h earlier than in the case of placebo (24.5 h vs. 42 h, respectively). Another important point to evaluate the effectiveness of baloxavir therapy was the median time to return to normal health, which was 129.2 h in the case of baloxavir and 168.8 h in the case of placebo, but this difference was not significant. On the other hand, clear differences were noted in terms of the median duration of virus detection, which was 24 h in the baloxavir group versus 72 h in the oseltamivir group and 96 h in the placebo group [17,134,142].

Another phase 3 study was CAPSTONE-2, a multicenter, double-blind, randomized study of baloxavir marboxil in participants with influenza at high risk of influenza complications. The study was performed analogously to CAPSTONE-1 in three groups with baloxavir, oseltamivir, or placebo treatment. A total of 2184 patients participated in the study. The CAPSTONE-2 results showed that the median time to improvement of influenza symptoms in the baloxavir group was significantly shorter than in the placebo group and similar to the oseltamivir group (73.2 h, 102.3 h, and 81.0 h, respectively). However, for influenza B infection, the median time to improvement of influenza symptoms in the baloxavir group was 27.1 h shorter than with oseltamivir. In addition, this time was significantly shorter in patients who started treatment in the first 12 h after symptom onset, and in the baloxavir group, it differed by 48 h on average compared to placebo. Treatment with baloxavir proved to be as effective as oseltamivir in terms of the time to fever resolution, averaging 30.8 h for baloxavir and 34.3 h for oseltamivir. Both drugs showed significant efficacy compared to placebo, with a median time to fever resolution of 50.7 h. However, in terms of the speed of recovery, no significant changes were found between the three groups (baloxavir 126.4 h, oseltamivir 126.9 h, and placebo 149.8 h). The influenza virus transmission data were similar to the CAPSTONE-1 results and showed that the decrease in virus titer in baloxavir-treated patients occurred much faster than in those treated with oseltamivir or placebo [143,145].

MiniSTONE2, a phase 3 multicenter, double-blind, randomized active (oseltamivir) controlled study, was conducted in children aged 1–12 years. The efficacy of influenza treatment with baloxavir was a secondary endpoint of this study. The results showed that the effectiveness of treating influenza in children was comparable between baloxavir and oseltamivir. The mean time to symptom relief was 138.1 h in the baloxavir group and 150.0 h in the oseltamivir group [140,144]. Phase 3 clinical trials are currently underway on the safety and efficacy of baloxavir marboxil in healthy pediatric participants (from newborn to 1 year old) with influenza-like symptoms. Completion of the research and publication of the results is planned for August 2022 [146].

The preventive efficacy of baloxavir in individuals exposed to influenza patients has also been investigated. Ikematsu et al. presented the results of a multicenter, double-blind, randomized, placebo-controlled clinical trial conducted in Japan during the 2018–2019 influenza season. In this study, 1.9% of participants using single-dose baloxavir presented with clinical symptoms of influenza confirmed by a laboratory test, compared to 13.6% of participants using placebo. Thus, baloxavir showed significant efficacy in preventing influenza after contact with a sick household member [147]. A retrospective study conducted by Umemura et al. analyzed the transmission of the influenza virus in home contacts and revealed that treatment of a flu patient with baloxavir vs. oseltamivir equally limited the source of infection for healthy household members [148].

## 4. Current Studies

The rapid evolution of the influenza virus, leading to a reduction in vaccine effectiveness and the emergence of drug-resistant strains, is driving continuous research into the development of new antiviral drugs. Currently, research is focused on several directions, including substances of synthetic, biological (bacterial), and plant origins.

One potential antiviral drug candidate is 1,3-dihydroxy-6-benzo [c] chromene (D715-2441). This is a small-molecule inhibitor that exhibits significant inhibitory activity against influenza A virus in vitro. Liu et al. showed that D715-2441 has antiviral activity against many types of influenza A virus, including H1N1, H3N2, H5N1, and H7N9, as well as oseltamivir-resistant strains with the H274Y NA mutation. The main target of the molecule is the early stage of viral replication. D715-2441 binds specifically to the PB2 protein, markedly inhibiting the activity of influenza RNA polymerase. In addition, simultaneous application of the tested molecule with zanamivir results in a synergistic antiviral effect [149].

Another promising compound is the novel compound FA-6005. In vitro studies have shown that this small molecule exhibits broad antiviral activity against human influenza A and B viruses. FA-6005 suppresses influenza virus replication and disrupts the intracellular transport of vRNP at many stages. This molecule interferes with various stages in the life cycle of influenza virus, including the processes of adsorption, entry, replication, transcription, and export. The molecular target of FA-6005 is amino acid residue 41, thereby inhibiting the activity of the vRNP complex. The interaction of FA-6005 has been shown to reduce NP/vRNP export, impair the trafficking of circulating RNPs in the cytoplasm, perturb the virus uncoating process and vRNP import, and lead to disruption of the budding of daughter virions [150].

Tests on bacterial RNA have also shown the inhibitory effect of FA-6005 on the influenza virus. In an in vivo study, heat-killed *Lactobacillus plantarum* SNK12 (SNK) and *Enterococcus faecalis* KH2 (KH2) strains were orally administered to mice infected with influenza A virus. In this study, Watanabe et al. showed that orally administered SNK and KH2 inhibited viral replication and increased the immune response [151].

Phytochemicals are also undergoing extensive research in the context of controlling influenza virus. One of them investigated the effect of catechin and gallic acid from *Toona sinensis* Roem leaves on influenza virus A (H1N1) in cell cultures. The study showed inhibition of viral replication and its adhesion to host cells by regulating cytokines and adhesion molecules (fractalkine, E-selectin, IL-8, VCAM-1, and ICAM-1) [152]. A follow-up to this study determined a 50% effective inhibitory concentration (EC50) of gallic acid and catechin for influenza A (H1N1) of 2.6 μg/mL and 18.4 μg/mL, respectively, and 50% cytotoxic concentration (CC50) of 22.1 μg/mL and >100 μg/mL for gallic acid and catechins, respectively. These results indicate that gallic acid is a sensitive substance for inhibiting influenza A (H1N1) infection and that catechin is a safe substance for long-term use in preventing influenza virus infection [153].

All of the above studies represent a promising step towards the development of new anti-influenza drugs. However, they need to be further investigated at the preclinical and clinical levels.

## 5. Discussion

Out of the three groups of anti-influenza drugs, only two are currently used in therapy and prophylaxis, NAI and CENI [8,49]. NAIs, including zanamivir, oseltamivir, peramivir, and laninamivir, are the most common group of drugs for influenza [37,38,39]. CENIs, on the other hand, are a completely new class of drugs with a unique representative recently available on the market, baloxavir marboxil [17,140]. The main difference between NAIs and CENIs is the mechanism of action [57].

Each of the drugs listed above has a unique feature that distinguishes it from the rest. Baloxavir stands out among the oral medications available in Europe for its single-dose therapy. This is its advantage over oseltamivir, which is by far the most widely used anti-influenza drug. Oseltamivir therapy requires a considerable regimen, as it lasts 5 days, and the capsules should be taken twice a day at 12 h intervals, which may cause a problem for some groups of patients, such as children [87,139]. Aside from baloxavir, laninamivir and peramivir are also single-dose medications. Such a dosing regimen is an ideal therapeutic solution in the face of the fight against drug resistance, which is favored by a break in treatment.

In the group of drugs for oral use, oseltamivir has the advantage of potential administration in the form of oral suspension, which is convenient for patients who have problems swallowing tablets or for whom the use of inhaled drugs poses an increased risk, such as patients with asthma [69]. In severe cases of influenza, in patients requiring hospitalization and sometimes extracorporeal membrane oxygenation (ECMO), intravenous infusion is often the best route of drug administration. This group of drugs includes zanamivir (Dectova^®^) and peramivir (Rapivap^®^), but only the former is approved for the treatment of seriously ill patients and life-threatening conditions [65,73].

All currently recommended antiviral drugs against influenza have a favorable safety profile. The most common AEs occurring with oseltamivir, intravenous zanamivir, peramivir, laninamivir, and baloxavir are headache and gastrointestinal disturbances, such as vomiting, nausea, and diarrhea [7,65,67,71,73,75,123]. These AEs usually accompany the first dose of the drug and disappear within the first 2 days of therapy. To alleviate these symptoms, it is recommended to take the drugs in the form of tablets or capsules with food [54,132]. The exception is zanamivir for inhalation (Relenza^®^), which is most often associated with disorders of the respiratory system [64]. NPAEs have been reported for all recommended anti-influenza medications. However, due to the fact that they come mainly from post-marketing surveillance and the specificity of flu symptoms, which are often associated with fever, it has not been possible to closely link these incidents to the use of any antiviral agent. Due to the serious nature of these symptoms, they require further attention and research [7,26,64,66,72].

The effectiveness of anti-influenza drugs has been the subject of many studies (Table 4). The key factor in increasing the effectiveness of therapy, regardless of the type of drug used, seems to be the time at which the drug is administered from the first symptoms of the disease. The best results were obtained when it was ≤12 h [111,129,145]. In this situation, the treatment period was shortened by up to 3 days compared to treatment 48 h after symptom onset. Another important factor seems to be the occurrence of fever. A better clinical response to treatment was observed in patients with fever compared to those without fever. In the case of zanamivir, the difference was 2 days on average (Table 4) [41,111]. The patient’s age may also be important, as the time to alleviation of symptoms (TTAS) has been reported to be shorter in children treated with oseltamivir than in adults [53,67].

As shown in Table 3, the use of antiviral drugs in influenza therapy has an effect in terms of shortening the TTAS from 0 to 3 days [41,111]. In the case of baloxavir, the difference is a little over 24 h regardless of the age group and is equivalent to the effectiveness of oseltamivir [142,143,144]. Furthermore, treatment with peramivir and intravenous zanamivir results in a therapeutic effect with TTAS and time to clinical stability comparable to those of treatment with oseltamivir [84,124,125]. This is important due to the possibility of intravenous administration of these drugs in hospitalized patients with severe disease for whom administration of oral oseltamivir may be problematic.

These results, though significant and noticeable due to the improvement in the patient’s comfort, do not seem to be satisfactory in achieving the optimal therapeutic benefit. On the other hand, the positive effect of zanamivir and oseltamivir on reducing complications was demonstrated [111,112]. Notably, laninamivir [128], zanamivir [66,78,86], and baloxavir [17,132] are among the drugs showing efficacy against some strains resistant to oseltamivir.

Despite an unsatisfactory treatment effect, the use of the above-described drugs in seasonal and post-exposure prophylaxis seems to be much more effective and amounts to about 80% efficacy for zanamivir [113,114,115], 70% to 90% for oseltamivir [12,14,53,58,118,119,120], and 62.8% for laninamivir. In the case of baloxavir, post-exposure cases occurred in 1.9% of the group using prophylaxis versus 13.6% in the placebo group [147]. However, it should be emphasized that vaccination remains the most effective type of flu prevention.

## 6. Conclusions

Treating human flu remains a challenge. The anti-influenza drugs available on the market seem to be insufficient in achieving an optimal therapeutic effect. Baloxavir, which is new on the market, exhibits activity against some oseltamivir-resistant strains and does not differ significantly in terms of safety and effectiveness from known drugs used recently for the treatment of influenza. However, its use in a wider group of society than is currently approved requires further research and is highly recommended. The presence of baloxavir on the market also brings new opportunities for combination therapy using drugs from two different groups in terms of the mechanism of action. No such studies have been performed yet, which indicates a new therapeutic direction.

## Figures and Tables

**Figure 1 ijerph-19-03018-f001:**
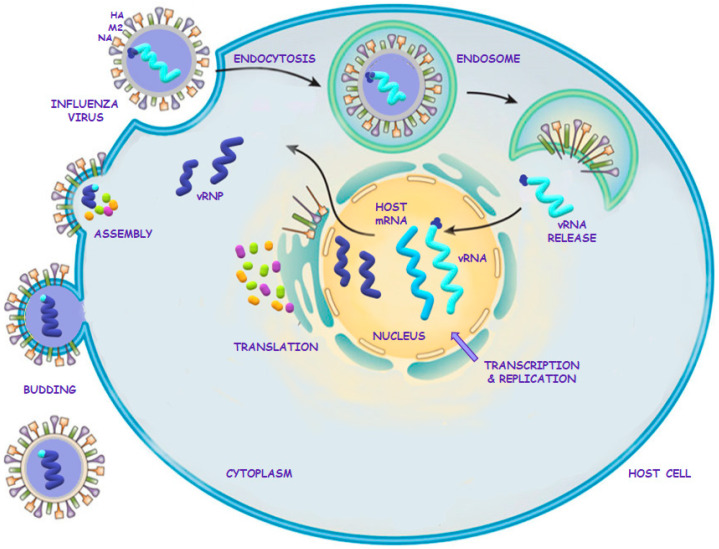
Life cycle of influenza A virus.

**Figure 2 ijerph-19-03018-f002:**
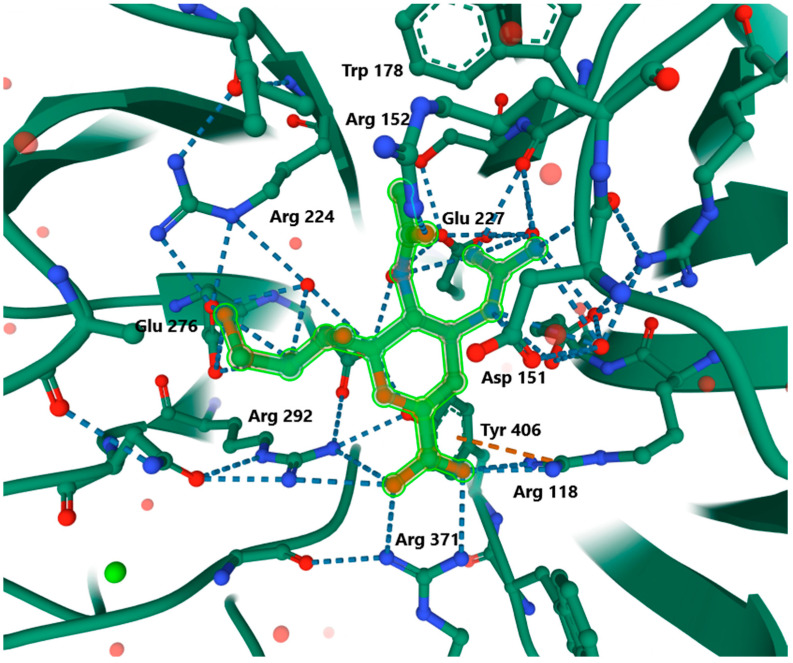
Influenza virus neuraminidase complexed with zanamivir.

**Figure 3 ijerph-19-03018-f003:**
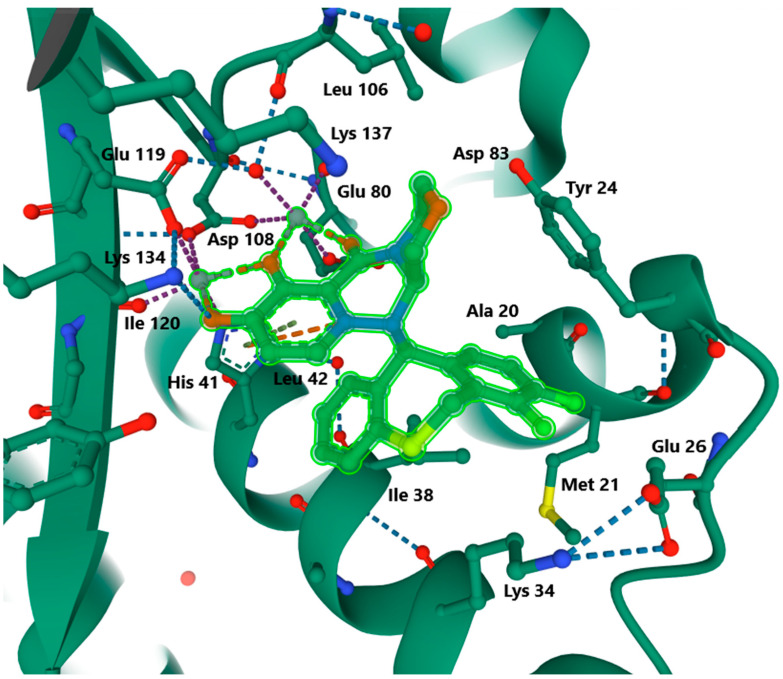
Influenza virus endonuclease complexed with baloxavir acid.

**Table 1 ijerph-19-03018-t001:** Antiviral drugs recommended for use against influenza viruses.

Drug Class	Active Substance	CID	Trade Name
NAI	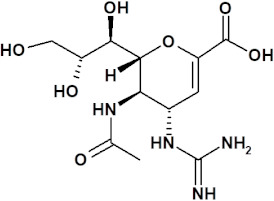 zanamivir	60855	Relenza® registered in EU, USA and AsiaDectova® registered in EU
NAI	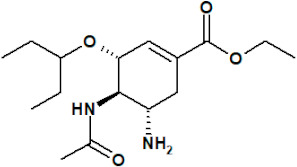 oseltamivir	65028	Tamiflu® registered in EU, USA and Asia
NAI	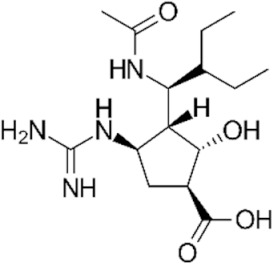 peramivir	154234	Rapivap® registered in USA and Asia
NAI	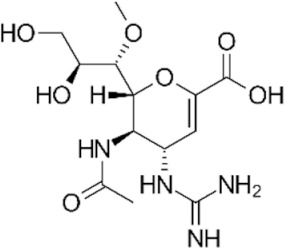 laninamivir	502272	Inavir® registered in Asia
CENI	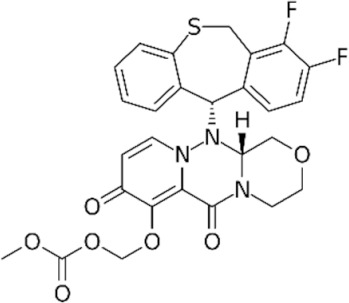 baloxavir marboxil	124081896	Xofluza® registered in EU, USA and Asia

**Table 2 ijerph-19-03018-t002:** Summary of frequency of serious or minor adverse effects associated with the administration of NAIs. Data from [9,12,14,16,43,59,62,64,65,66,67,68,69,70,71,72,73,74,75,76].

Drug	Frequency	Adverse Effects
Zanamivir for oral inhalation	1% to 10%	Skin reaction, such as rash
	0.1% to 1%	Disordered respiratory function, bronchospasm, throat tightness or constriction, vasovagal-like reactions, and allergic-type reactions, including oropharyngeal edema and urticaria.
	<0.01%	Anaphylactic reaction, facial edema, toxic epidermal necrolysis, erythema multiforme, and Stevens–Johnson syndrome
	Underestimated	Neuropsychiatric adverse effects (NPAEs), seizures, delirium, hallucination, abnormal behavior, and depressed level of consciousness
Intravenous zanamivir	1% to 10%	Diarrhea, rash, hepatocellular injury, increased levels of transaminases (ALT and AST), neutropenia, and renal failure
	0.1% to 10%	Urticaria and increased alkaline phosphatase
	Underestimated	Anaphylactic reaction, facial and oropharyngeal edema, NPAEs, renal impairment, paralytic ileus, and hypotension
Oseltamivir	>10%	Headache and nausea
	1% to 10%	Vomiting, bronchitis, sore, throat, nasopharyngitis, sinusitis, pain, and dizziness
	0.1% to 1%	Hypersensitivity reaction, rash, urticaria, dermatitis, cardiac arrhythmia, and convulsions
	00.1% to 0.1%	Thrombocytopenia, anaphylactic reactions, toxic epidermal necrolysis, hepatic failure, hepatitis, evaluated liver enzymes, gastrointestinal bleeding, visual disturbances, and NPAEs
Peramivir	>10%	Diarrhea
	1% to 10%	Neutropenia, nausea, vomiting, injection site rash, and increased AST and ALT
	Underestimated	Insomnia, fever, proteinuria, tympanic membrane erythema, anaphylactic reactions, severe dermatological reactions such as Stevens–Johnson syndrome and exfoliative dermatitis, and NPAEs
Laninamivir	1% to 10%	Cough, diarrhea, and headache
	0.1% to 1%	Gastritis, abnormal behavior, and nervous system disorders

**Table 3 ijerph-19-03018-t003:** Indication and dosage of anti-influenza medications.

Drug	Therapeutic Indication	Age Interval	Pharmaceutical Form	Dose
Zanamivir	Treatment of acute uncomplicated influenza A and B	≥5 years and older (≥7 years in USA and Canada)	Powder for oral inhalation	10 mg twice daily for 5 days
	Post-exposure prophylaxis	≥5 years and older		10 mg once daily for 10 days
	Seasonal prophylaxis	(≥7 years in Canada)		10 mg once daily for 28 days
Zanamivir	Treatment of hospitalized seriously ill patients	≥6 months and older	Solution for infusion	Weight-based dose 6 months < 6 years 14 mg per kg 2xd ≥ 6 < 18 years 12 mg per kg 2xd Adults > 50 kg 600 md 2xd for 5–10 days
Oseltamivir	Treatment of acute uncomplicated influenza A and B	No age limits (EU) ≥2 weeks and older (USA)	Capsules 30 mg, 45 mg, 75 mg Oral suspension	≥13 years: 75 mg 2xd for 5 days <13 years old—weight-based dose
	Post-exposure prophylaxis	≥1 year old		≥13 years: 75 mg 1xd for 10 days at least
	Seasonal prophylaxis	No age limits (EU)		75 mg 1xd for up to 6 weeks
Peramivir	Treatment of acute uncomplicated influenza A and B	≥18 years old	Solution for infusion	2 years ≤ 12 years single 12 mg/kg
				Adults: single 600 mg i.v.
Laninamivir	Treatment of acute uncomplicated influenza A and B Post-exposure prophylaxis	No age limits	Oral inhalation powder	<10 years 20 mg single dose ≥10 years 40 mg single <10 years 20 mg single dose ≥10 years 40 mg single dose
Baloxavir marboxil	Treatment of acute uncomplicated influenza A and B (healthy and high risk of complications in USA)	≥12 years old	Tablets for oral use	Weight-based dose 40 > 80 kg 40 mg single dose ≥80 kg 80 mg single dose
	Post-exposure prophylaxis (in EU)			

**Table 4 ijerph-19-03018-t004:** The effectiveness of anti-influenza drugs.

Type of Treatment	Primary End Point	Effect	Study
Zanamivir vs. placebo given up to 30 h of symptom onset	TTAS *	1 day shorter with zanamivir	Hayden et al. [41]
Zanamivir vs. placebo (patients with fever) given up to 30 h of symptom onset	TTAS	3 days shorter with zanamivir	Hayden et al. [41]
Zanamivir vs. placebo	TTAS	0.6 day (14.4 h) shorter with zanamivir	Heneghan et al. [53]
Zanamivir vs. placebo (patients without fever)	TTAS	0 (no significant differences)	The MIST Study Group [111]
Zanamivir vs. placebo (patients with fever)	TTAS	2 days shorter with zanamivir	The MIST Study Group [111]
Zanamivir vs. placebo (high-risk patients)	TTAS	2.5 days shorter with zanamivir	The MIST Study Group [111]
Dectova vs. oseltamivir (patients hospitalized in serious condition and ICU ** patients)	TTAS	Similar effect	Marty et al. [84]
Oseltamivir vs. placebo (adult patients with fever)	TTAS	1.3 days shorter with oseltamivir	Tamiflu summary of product characteristics [67]
Oseltamivir vs. placebo (pediatric patients)	TTAS	1.5 days shorter with oseltamivir	Tamiflu summary of product characteristics [67]
Peramivir 200 mg or 400 mg vs. oseltamivir (hospitalized patients)	Time to clinical stability	P200mg—31.0 h P400mg—24.3 h Oseltamivir—35.5 h	Ison et al. [125]
Peramivir 300 mg or 600 mg vs. oseltamivir	TTAS	P300mg—78.0 h P600mg—81.0 h Placebo—81.8 h	Kohno et al. [124]
Peramivir vs. oseltamivir (pediatric patients)	Fever duration	A significant advantage of peramivir	Shobugawa et al. [126]
Peramivir vs. zanamivir inhalation (pediatric patients)	Fever duration	Peramivir 1 day shorter than zanamivir	Hikita et al. [127]
Peramivir vs. laninamivir	Fever duration	Peramivir 1 day shorter than laninamivir	Hikita et al. [127]
Laninamivir 20 mg or 40 mg vs. oseltamivir	TTAS	L20mg—85.8 h L40—73.0 h Os—73.6 h	Watanabe et al. [128]
Baloxavir vs. placebo (adolescents)	TTAS	Baloxavir—53.7 h Placebo—80.2 h	CAPSTONE-1 [142]
Baloxavir vs. oseltamivir vs. placebo (adult patients)	TTAS	Baloxavir—53.7 h Oseltamivir—53.8 h Placebo—80.0 h	CAPSTONE-1 [142]
Baloxavir vs. placebo	Fever duration	Baloxavir—24.5 h Placebo—42 h	CAPSTONE-1 [142]
Baloxavir vs. oseltamivir vs. placebo	TTAS	Baloxavir—73.2 h Oseltamivir—81.0 h Placebo—102.3 h	CAPSTONE-2 [143]
Baloxavir vs. oseltamivir (pediatric patients)	TTAS	Baloxavir—138.1 h Oseltamivir—150.0 h	MiniSTONE [144]

* TTAS—time to alleviation of symptoms; ** ICU—intensive care unit.

## Data Availability

Not applicable.

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
