# Peer review of "Antiviral Drugs in Influenza"

_ijerph, 2022, doi:10.3390/ijerph19053018_

Round 1

Reviewer 1 Report

The manuscript entitled “Antiviral drugs in influenza” describes the available information on several drugs used for the treatment of influenza and their effectiveness. The authors have nicely summarized the therapeutic indications, dosage, and effectiveness of antiviral drugs in the form of Tables. The manuscript lacks the structural and molecular details on the mode of action of antivirals. The manuscript also lacks schematic representations/figures. Once authors effectively organize the manuscript, it should be a good resource for the general audience interested in recent developments in antiviral treatments. Overall, the manuscript is poorly written.

I recommend authors to work on the following comments to improve the quality of the manuscript.

Major comments:

  1. Authors should include a schematic representation showing the influenza virus life cycle and where this different class of drugs act. It will help the audience to understand the flow of the content in a better way.
  2. Structures are available for most Influenza drugs. In my opinion, authors should have one figure that includes 2D structures with their PubChem ID for all the described drugs.
  3. The authors have provided a little information on Neuraminidase structure (Point 2.2.1.) in the text without showing its structure which makes it difficult to understand for a general audience. The authors should include a figure for point 2.2.1. NAIs mode of action. There are a few structures of Neuraminidase bound to its inhibitors that are available in “Protein Data Bank (PDB)”.
  4. Authors should focus on the detailed mechanism of action, available pharmacokinetics, and structural insights together with effectiveness (already included) instead of describing very basic information which has already been reviewed in an effective manner and provided in product monographs of these drugs. I recommend authors carefully reorganize the manuscript.

Minor comments:

  1. Line 58, write “intravenous”.
  2. Line 59, Correct to “The first of them is used as antiviral treatment”.
  3. Title of the sub-point 2.1.1. is not appropriate. In my opinion, it should be “Other uses for Amantadine”.
  4. Line 77-78, (Recently……ADS5102) Sentence isn’t clear. Please rewrite it correctly.
  5. Line 82, put a full stop.
  6. Throughout the manuscript, terms “In vitro” and “in vivo” should be in italics.
  7. Line 113-115, correct to “The currently available NAIs for general use include zanamivir,
  8. oseltamivir, peramivir, and laninamivir [34,35,36].” I think, there is no need to mention the route of administration.
  9. Line 125, correct it to “…hospitalization time period in seriously…”
  10. Line 127, correct it to “…confirmed influenza infection, it…”
  11. Line 156, please remove the following sentence “This structure of zanamivir determines its administration by oral inhalation [34].”
  12. Line 183, correct it to “Zanamivir for oral inhalation”.
  13. Line 184, correct it to “…(GSK) is zanamivir…”.
  14. Line 196, Correct it to “zanamivir for inhalation is licensed for…”
  15. Line 218, correct it to “A contraindication to the use of zanamivir for oral inhalation is an allergy to milk protein…”
  16. Line 221-223, please delete the following sentence: “GlaxoSmithKline does not provide any other form of drug administration than the one indicated in the product characteristics, i.e. using a specially designed inhaler called Diskhaler.”
  17. Line 234, Please do not write “Inhaled Zanamivir”. Correct it to “Zanamivir for oral inhalation”. Please make this correction throughout the manuscript.
  18. Authors shouldn’t use the abbreviation for “adverse effects” throughout the manuscript.
  19. Line 256, correct it to “…does not allow the use of medications suitable for oral administration or inhalation, such as patients with…”
  20. Line 448, please delete “it”.
  21. Line 515-519, Sentence is difficult to understand. Please simplify and rewrite to make it clearer.
  22. The manuscript has two “Table 2”. Please make the correction.

Author Response

Dear Reviewer,

Thank you very much for reviewing our manuscript. We considered all your comments and introduced required changes to significantly improved the value of our paper.

With reference to your suggestions, below we present our changes in detail.

  1. Authors should include a schematic representation showing the influenza virus life cycle and where this different class of drugs act. It will help the audience to understand the flow of the content in a better way.
    In section 2.2. Influenza virus life cycle, we have described and included a figure (Figure 1.) showing the life cycle of the influenza virus. Both in the description and in the figure, the sites for the action of all the drugs presented in the manuscript are shown.

  2. Structures are available for most Influenza drugs. In my opinion, authors should have one figure that includes 2D structures with their PubChem ID for all the described drugs.

    The drug structures with their PubChem ID are presented in Table 1. Antiviral drugs recommended for use against influenza viruses, classification and structure of compounds

  3. The authors have provided a little information on Neuraminidase structure (Point 2.2.1.) in the text without showing its structure which makes it difficult to understand for a general audience. The authors should include a figure for point 2.2.1. NAIs mode of action. There are a few structures of Neuraminidase bound to its inhibitors that are available in “Protein Data Bank (PDB)”.

    In section 3.2.2. NAIs mode of action, we described the structure of neuraminidase in detail and added Figure 2. Influenza virus neuraminidase complexed with zanamivir.

  4. Authors should focus on the detailed mechanism of action, available pharmacokinetics, and structural insights together with effectiveness (already included) instead of describing very basic information which has already been reviewed in an effective manner and provided in product monographs of these drugs. I recommend authors carefully reorganize the manuscript.

The manuscript has been carefully reorganized. The structure and life cycle of the influenza virus was presented, a detailed description of mode of action and the available pharmacokinetics was added. The structure of drugs and their complex with proteins of the influenza virus at active sites were also included according to your instructions.

  1. All of your minor comments have been included and corrected in the manuscript.

We would be most grateful if you reconsider recommendation for publication.

Yours faithfully,
Authors

Reviewer 2 Report

This is a good start to a review on a topic that is very necessary, however, I think you need to relook at your topic organization and group things slightly better and add some items to your paper to make it stand out vs others that do similar reviews. 

Introduction

  1. Line 31 - you can remove the comma after rate in "...an annual global influenza attack rate ranges from..."
  2. Lines 46 - 49 - perhaps rephrase this so that it sounds more "scientific"
  3. To provide some context to the workings of the medications, you should add some description of the life cycle of influenza so that readers get context for the workings of these medications - a diagram may also be helpful

Drugs used for influenza treatment and prophylaxis

  1. I'm not sure you need to spend that much time talking about other indications for amantadine - you spend more time talking about its other uses than you do talking about influenza, which is the focus of the paper. I would recommend that you remove that section since it is not relevant; you can leave a few sentences about its use in COVID since it is a "hot topic" but do not spend too much time on it. 
  2. I would also talk about treatment doses and duration for amantadine
  3. Lines 115 - 117 - you state that "All of them are effective against most influenza A and B strains" - what do you mean by effective? You need to explain this and provide actual data and numbers. What about safety?

General:

  1. Are there any experimental or new treatments currently being researched? What is the data for these experimental or new agents? There are tons of review articles relating to the treatment of influenza, I would recommend that you add something like that so that it makes your article stand out. 
  2. Consider the addition of a table that lists the medications, their dosages, their uses, and the effectiveness and safety (common adverse reactions) as a type of summary for your paper 
  3. Perhaps you have sections that (1) look at the MOA of each medication (2) look at the effectiveness of each medication (3) compare the effectiveness of each medication instead of having them under each drug. This will ensure that you have the same level of detail for each of the medications.

Author Response

Dear Reviewer,

Thank you very much for reviewing our manuscript and for valuable comments and suggestions.

With reference to your suggestions, below we present our changes in detail.

Introduction

  1. Line 31 - you can remove the comma after rate in "...an annual global influenza attack rate ranges from..."

    The comma has been removed.

  1. Lines 46 - 49 - perhaps rephrase this so that it sounds more "scientific"

    The sentence has been corrected.

  2. To provide some context to the workings of the medications, you should add some description of the life cycle of influenza so that readers get context for the workings of these medications - a diagram may also be helpful

    The life cycle of influenza virus with a diagram was added in section 2.2.

Drugs used for influenza treatment and prophylaxis

  1. I'm not sure you need to spend that much time talking about other indications for amantadine - you spend more time talking about its other uses than you do talking about influenza, which is the focus of the paper. I would recommend that you remove that section since it is not relevant; you can leave a few sentences about its use in COVID since it is a "hot topic" but do not spend too much time on it.

    Paragraph 2.1.1. (the other application of amantadine) has been removed and a brief reference to the use of amantadine in Sars Cov-2 therapy has been included in section 3.1. Amantadine.

  2. I would also talk about treatment doses and duration for amantadine

    We intentionally did not include information on the amantadine treatment protocol for treating influenza as it is not currently recommended as an anti-influenza drug. We have listed amantadine among all drugs mainly in historical context.

  3. Lines 115 - 117 - you state that "All of them are effective against most influenza A and B strains" - what do you mean by effective? You need to explain this and provide actual data and numbers. What about safety?

    The effectiveness of NAIs is detailed and compared in section 3.2.7. Comparison of the effectiveness of NAIs’ and in Table 4. The effectiveness of anti-influenza drugs. On the other hand, the safety of drugs from this group has been described for each drug separately and for better imaging is presented in Table 2. Frequency of patients with serious or minor adverse effects associated with the administration of NAIs.

General:

  1. Are there any experimental or new treatments currently being researched? What is the data for these experimental or new agents? There are tons of review articles relating to the treatment of influenza, I would recommend that you add something like that so that it makes your article stand out.

    Following a very valuable advice, we have added section 4. Current studies, where the ongoing research on potential candidates for new anti-influenza drugs were presented.

  2. Consider the addition of a table that lists the medications, their dosages, their uses, and the effectiveness and safety (common adverse reactions) as a type of summary for your paper

    The following tales have been included in the manuscript:
    - Table 2. Frequency of patients with serious or minor adverse effects associated with the administration of NAIs.
    - Table 3. Indication and dosage of anti-influenza medications.
    - Table 4. The effectiveness of anti-influenza drugs

  3. Perhaps you have sections that (1) look at the MOA of each medication (2) look at the effectiveness of each medication (3) compare the effectiveness of each medication instead of having them under each drug. This will ensure that you have the same level of detail for each of the medications.

    For better readability of the text, the manuscript has been divided into sections: mode of action, pharmacokinetics, treatment and prophylaxis, and for a better illustration of the safety of individual drugs, it has been compared in Table 2. Frequency of patients with serious or minor adverse effects associated with the administration of NAIs.                                    We would be most grateful if you reevaluate our manuscript.

Yours faithfully,
Authors

Reviewer 3 Report

The manuscript provide an detailed overview regarding antiviral agents used for treatment of influenza virue type A and B

In particular the use of neuraminidase inhibitors (NAIs) and endonuclease inhibitors. several information including indication, dosage and side effects are provided considering the literature from clinical studies and documents from regulatory authorities.

The manuscript could be of great interest in pharmacy and medical fields and consequently I recommend the publication.

Nerveless some change is required, in my opinion, in order to improve the manuscript.

1) The paragraph 2.1.1 (other application of amantadine) is not necessary. Relevant information important for the use of this drug as antiviral (for example for Sars-Cov-2 treatment should be resumed and included in paragraph 2.1.

2) In paragraph 2.2, the biological role of viral NA (during infection or other relevant viral processes) should be shortly described as well as the relevance of this enzyme as an antiviral target.

3) A figure including the structure of NAIs considered in the manuscript could be useful.

4) In many parts of the manuscript several not relevant details are provided and this make difficult the reading of the manuscript

For example:

In line 191 you report that lactose contain milk protein. This is relevant? Why?

In lines 396-400 details concerning the preparation of the solution could be omitted

Many other part should be reduced avoiding not relevant technical dettails

5) Please report the reaction of metabolic activation of Baloxavir

6) the discussion should be reduced. Many details have been already reported in the previous paragraphs. These parts should be resumed providing only the essential information.

Author Response

Dear Reviewer,

Thank you very much for reviewing our manuscript. We introduced your suggestions to improve the quality of our paper.
With reference to your comments, below we present our changes in detail.

1) The paragraph 2.1.1 (other application of amantadine) is not necessary. Relevant information important for the use of this drug as antiviral (for example for Sars-Cov-2 treatment should be resumed and included in paragraph 2.1.

Paragraph 2.1.1. (the other application of amantadine) has been removed and information about attempts to use amantadine treatment in Sars-Cov-2 therapy has been included in section 3.1. Amantadine comprehensively describing this drug.

2) In paragraph 2.2, the biological role of viral NA (during infection or other relevant viral processes) should be shortly described as well as the relevance of this enzyme as an antiviral target.

In section 3.2.2. NAIs mode of action, we described the structure, function and method of NA binding to a substrate (a drug from the NAIs group).

3) A figure including the structure of NAIs considered in the manuscript could be useful

In Table 1, we present the structures of all the drugs NAIs and CENI mentioned by us in the manuscript.

4) In many parts of the manuscript several not relevant details are provided and this make difficult the reading of the manuscript.

We have removed all the text fragments proposed by you and many others.

5) Please report the reaction of metabolic activation of Baloxavir

In paragraph 3.3.3. Pharmacokinetics of Baloxavir we presented the metabolic activation of Baloxavir.

6) the discussion should be reduced. Many details have been already reported in the previous paragraphs. These parts should be resumed providing only the essential information.

According to your advice, the discussion has been significantly shortened.

We hope that in present form you find the manuscript acceptable for publication.

Yours faithfully,
Authors

Round 2

Reviewer 1 Report

I have the following minor comment.

In Figures 2 and 3, protein and drug molecules should be colored differently.

Author Response

Dear Reviewer,

thank you very much for your comment. We have changed the colours in Figure 2 and 3.

Yours sincerely,

Authors

Reviewer 2 Report

Thank you for the updates!

Author Response

Dear Reviewer,

thank you!

Yours sincerely,

Authors